# 8-Week Supplementation of 2S-Hesperidin Modulates Antioxidant and Inflammatory Status after Exercise until Exhaustion in Amateur Cyclists

**DOI:** 10.3390/antiox10030432

**Published:** 2021-03-11

**Authors:** Francisco Javier Martínez-Noguera, Cristian Marín-Pagán, Jorge Carlos-Vivas, Pedro E. Alcaraz

**Affiliations:** 1Research Center for High Performance Sport, Catholic University of Murcia, Campus de los Jerónimos Nº 135, UCAM, 30107 Murcia, Spain; cmarin@ucam.edu (C.M.-P.); palcaraz@ucam.edu (P.E.A.); 2Health, Economy, Motricity and Education Research Group (HEME), Faculty of Sport Sciences, University of Extremadura, Avda. de Elvas, s/n., 06006 Badajoz, Spain; jorge.carlosvivas@gmail.com

**Keywords:** polyphenols, flavonids, endogenous antioxidant enzymes, reduced glutathione, oxidized glutathione, catalase, superoxide dismutase, interleukin 6, tumor necrosis factor, endurance sports

## Abstract

Both acute and chronic ingestion of 2S-hesperidin have shown antioxidant and anti-inflammatory effects in animal studies, but so far, no one has studied this effect of chronic ingestion in humans. The main objective was to evaluate whether an 8-week intake of 2S-hesperidin had the ability to modulate antioxidant-oxidant and inflammatory status in amateur cyclists. A parallel, randomized, double-blind, placebo-controlled trial study was carried out with two groups (500 mg/d 2S-hesperidin; *n* = 20 and 500 mg/d placebo; *n* = 20). An incremental test was performed to determine the working zones in a rectangular test, which was used to analyze for changes in antioxidant and inflammatory biomarkers. After 2S-hesperidin ingestion, we found in the rectangular test: (1) an increase in superoxide dismutase (SOD) after the exercise phase until exhaustion (*p* = 0.045) and the acute recovery phase (*p* = 0.004), (2) a decrease in the area under the oxidized glutathione curve (GSSG) (*p* = 0.016), and (3) a decrease in monocyte chemoattractant protein 1 (MCP1) after the acute recovery phase (*p* = 0.004), post-intervention. Chronic 2S-hesperidin supplementation increased endogenous antioxidant capacity (↑SOD) after maximal effort and decreased oxidative stress (↓AUC-GSSG) during the rectangular test, decreasing inflammation (↓MCP1) after the acute recovery phase.

## 1. Introduction

Flavonoids are bioactive substances found mainly in fruits and vegetables, with more than 15,000 molecules identified within this family [1]. However, one of the most well-known is hesperidin, which is a flavonoid present at high concentrations in citrus fruits, being the main one in sweet orange (*Citrus sinensis*). Hesperidin may be found in two isomeric forms, 2S- and 2R-, where the 2S isomer is predominant in nature [2]. When hesperidin reaches the intestine, bacterial flora converts it into hesperetin (aglycon), which is effectively absorbed, being the main metabolite of hesperidin [3]. Previous studies have shown the positive effects of hesperidin on some diseases (neurological, cardiovascular, insulin sensitization) due to its antioxidant and anti-inflammatory properties [4,5]. Moreover, the intake of hesperidin (in orange juice) has been shown to modulate leukocyte gene expression, boosting its antioxidant and inflammatory profile, and therefore showing a nutrigenomic effect [6]. On the other hand, the ability of 2S-hesperidin to improve performance has been observed [7]. It should be noted that there are other important factors that can modulate the effect of flavonoids like hesperidin, such as intestinal flora transformations, absorption and bioavailability [8].

The antioxidant effect of hesperidin is mainly related to its radical scavenging capabilities, as well as the increase in antioxidant cellular defense catalase (CAT), superoxide dismutase (SOD), reduced glutathione (GSH) and oxidized glutathione (GSSG) via the nuclear respiratory factor 2 (NRF2) signaling pathway [4]. On the other hand, the hesperidin anti-inflammatory effect is produced by a decrease in inflammatory markers, such as nuclear factor kappa B (NF-κB), interleukin 6 (IL6), tumor necrosis α (TNFα) and inducible nitric oxide synthase (iNOS) [4].

Regarding the potential of hesperidin on physical performance, a recent study reported that the acute intake of 500 mg of 2S-hesperidin significantly improved anaerobic performance [9]. In the same study, they also found small non-significant changes in CAT, SOD, GSH and the GSSG/GSH ratio compared to a placebo during a rectangular test (with different intensities) in amateur cyclists. Similarly, a study performed in rats observed that 2S-hesperidin (200 mg/kg, three days per week during five weeks) showed a protective effect on the oxidative stress induced by an exhausting exercise [10]. Hesperidin supplementation prevented the increase in reactive oxygen species (ROS) production and avoided a decrease in SOD and catalase activities, while leading to a higher physical performance. In the same way, 6 weeks of hesperetin (main metabolite of hesperidin) supplementation (50 mg·kg^−1^·d^−1^) significantly increased the GSH/GSSG ratio and improved running performance (exercise time) in aged mice [11]. In addition, a recent study found that eight weeks’ intake of 2S-hesperidin improved performance at the threshold of estimated functional power and maximum power in an incremental test until exhaustion compared to a placebo in amateur cyclists [7]. Other polyphenols have shown hesperidin-like effects. For example, the intake of 100 mL per day for six weeks of acai berry-based juice (↑ anthocyanins) increased the levels of GSH and CAT post-exercise and after 1 h of recovery, without changes in SOD and exercise performance (300 m running times) in junior athletes [12]. 

With regards to exercise, it is known that almost 0.15% of the oxygen consumed is converted into ROS, which can be detrimental to muscle and mitochondrial function [13]. In sports physiology, it is hypothesized that rapid increases in ROS during intensive exercise may be a contributor to fatigue [14]. Based on recent findings, a new theory proposes that antioxidant supplementation (vitamins A, C, E, thiols, ubiquinones and flavonoids) may delay fatigue [15]. However, this mitigation of ROS generation may disrupt cellular signaling involved in training adaptations [16]. ROS are intracellular messengers and activators of transcription factors that promote the expression of genes related to training adaptations and performance improvement [16]. Thus, antioxidant supplementation could decrease ROS production and delay fatigue, but in turn it may slow down the physiological adaptations of training [17,18]. Due to the current controversy on this topic, further investigations are required to evaluate if the intake of antioxidant polyphenols, such as hesperidin, could improve endogenous antioxidant status without negatively affecting performance. 

Currently, studies have shown that acute [9] and chronic [7] intake of 2S-hesperidin in amateur cyclists improve anaerobic and aerobic performance, respectively. However, no research has explained the metabolic, biochemical and molecular mechanisms by which 2S-hesperidin intake improves performance. We hypothesized that the chronic intake of 2S-hesperidin would improve amateur cyclists’ antioxidant status, evaluated through markers such as CAT, SOD, GSSG, GSH and hemoxigenase 1(HO1), but decrease inflammatory markers, such as IL6, TNFα, monocyte chemoattractant protein-1 (MCP1) and C reactive protein (CRP). However, the implications of long-term or prolonged use are unknown. Therefore, this study aimed to evaluate the effect of eight weeks of 2S-hesperidin supplementation (500 mg/day) on the antioxidant-oxidant (CAT, SOD, GSH, GSSG, HO1 and TBARS) and anti-inflammatory (IL6, TNFα, MCP1 and CRP) state in amateur cyclists before and at the end of the rectangular test and after the resting phase.

## 2. Methodology

### 2.1. Study Design

A randomized, double-blind, parallel clinical trial was conducted. Forty subjects were divided into 2 groups: 2S-heperidin (*n* = 20) and placebo (*n* = 20). Subjects were randomized into groups using the Randomizer software. Participants consumed two 250 mg capsules of either Placebo (microcellulose, 500 mg) or 2S-hesperidin (500 mg Cardiose^®^, produced by HealthTech BioActives (HTBA), Murcia, Spain) at breakfast for 8 weeks. The Cardiose^®^ supplement consisted of a natural orange extract that, due to its unique manufacturing process, retains most of the natural isomeric form of hesperidin (NLT 85% 2S-hesperidin). The placebo supplements were similar in appearance to the 2S-hesperidin capsule. Cyclists were instructed to continue their usual diet and training program. The usual total training distance was balanced between the groups (Table 1).

### 2.2. Participants

Forty healthy male, amateur cyclists completed the study (Table 1). Subjects met the following inclusion criteria: 18–55 years old, BMI of 19–25.5 kg·m^−2^, at least 3 years of cycling experience, and training for 6–12 h·wk^−1^. Amateur cyclists were excluded if: (a) regular smoking or alcohol drinking, (b) metabolic, cardiorespiratory or digestive pathology or abnormality, (c) injury in the previous 6 months, (d) supplements or medication in the previous 2 weeks and (e) abnormal values in blood test parameters. Before the start of the study, participants were informed about the procedures, and signed informed consent was obtained. The study was conducted following the Declaration of Helsinki guidelines for research on human subjects [19] and was approved by the Ethics Committee of the Catholic University of Murcia (CE091802), registered in ClinicalTrials.gov (Identifier: NCT04597983).

### 2.3. Procedures

Participants visited the laboratory on five different occasions. Visit 1 consisted of a medical examination and blood extraction to determine health status. On visits 2 and 4, a 24-h diet recall was conducted, followed by an incremental test until exhaustion on a cycle ergometer to estimate the rectangular test zones. On visits 3 and 5, the 24-h diet recall was repeated, and participants performed a rectangular test on the cycle ergometer (Figure 1) (Table 2). Before each testing session (visits 2, 3, 4 and 5), a standardized breakfast composed of 95.2 g of carbohydrates (68%), 18.9 g of protein (14%) and 11.3 g of lipids (18%) was prescribed by a sports nutritionist. Intake of both treatments began at visit 1 under the supervision of an investigator and finished at visit 5. Subjects in both groups were instructed not to consume foods with a high content of citrus flavonoids (grapefruit, lemons, or oranges) for 5 days prior to and during the study. This was verified by diet recall records.

### 2.4. Testing

#### 2.4.1. Medical Exam

A medical examination was conducted by the research center’s medical doctor and consisted of medical and health history, resting electrocardiogram and examination (auscultation, blood pressure, etc.). These evaluations confirmed that the volunteer was healthy enough to be enrolled in the study.

#### 2.4.2. Maximal Test

Incremental step with final ramp test was performed on a cycle ergometer using a metabolic cart (Metalyzer 3B. Leipzig, Germany) to determine the maximal fat oxidation zone (FatMax), ventilatory thresholds 1 (VT1) and 2 (VT2) and maximal oxygen consumption (VO_2max_). Participants started cycling at 35 W for 2 min, increasing by 35 W for every 2 min until RER > 1.05, and then initiating the final ramp (+ 35 W·min^−1^) until exhaustion. To ensure they reached VO_2max_, at least 2 of the following criteria had to be fulfilled: plateau in final VO_2_ values (increase ≤ 2.0 mL·kg^−1^·min^−1^ in the 2 last loads), reaching maximal theoretical HR ((220–age)·0.95), RER ≥ 1.15 and lactate ≥ 8.0 mmoL/L. Ventilatory thresholds were obtained using the ventilatory equivalents method described by Wasserman [20].

#### 2.4.3. Rectangular Test

Rectangular test procedures are shown in Figure 1. This test was performed on a cycle ergometer using power output values achieved during the maximal test at different intensity zones (FatMax, VT1, VT2 and maximum power). Participants exercised continuously as follows: 10 min at FatMax, 10 min at VT1, 10 min at VT2, at maximum power until exhaustion (post-P_MAX_) and 30 min rest (post-REC). There were no rest periods between phases.

#### 2.4.4. Blood Samples

Venous blood (arm antecubital area) was collected into one 3 mL ethylenediaminetetraacetic acid (EDTA) tube for hemogram and another 3.5 mL polyethene terephthalate (PET) tube was collected by a nurse for overall health analysis (visit 1). Red blood cell count was carried out in an automated Cell-Dyn 3700 analyzer (Abbott Diagnostics, Chicago, IL, USA) using internal (Cell-Dyn 22) and external (Program of Excellence for Medical Laboratories-PEML) controls. Values of erythrocytes, haemoglobin, haematocrit and haematimetric indexes were estimated.

Additionally, venous blood samples were collected in the baseline, after the maximum power stage (post-P_MAX_) and during the resting phase (post-REC), to measure antioxidant and anti-inflammatory parameters (visits 3 and 5) (Figure 1). During every extraction point, 6 tubes of 3 mL of EDTA were obtained. Blood samples were centrifuged at 3500 rpm in 4 °C for 10 min and sent to the laboratory for later analysis.

#### 2.4.5. Urine Samples

The main hesperidin metabolites were analyzed in the urine of participants. Urine samples were collected for 24 h before V2 and V5 visits from each participant, before and after the supplementation, and were frozen in liquid nitrogen after collection and thawed for its analysis. For analysis, 50 µL of urine was mixed with 100 µL of water with 1% formic acid containing the internal standard. Then, the mixture was injected into LC-MS/MS (UHPLC 1290 Infinity II Series coupled to a QqQ/MS 6490 Series Agilent Technologies, Sta. Clara, CA, USA). Metabolites were quantified by external standard calibration using *rac*-Hesperetin-d3 as the internal standard.

### 2.5. Antioxidant and Inflammatory State Markers

The following parameters were selected to measure the antioxidant and inflammatory status.

#### 2.5.1. TBARS (Lipoperoxidation Biomarker)

Thiobarbituric acid reactive substances (TBARS) are a by-product of the oxidative degradation of lipids by reactive oxygen species (lipid peroxidation), which is commonly used as an oxidative stress marker [21]. TBARS assay involves the reaction of malondialdehyde (MDA), a product of lipid peroxidation, with thiobarbituric acid (TBA) under high temperature and acidic conditions to form an MDA-TBA complex that can be measured colorimetrically [22]. The coefficient of variation between replicas had to be less than or equal to 4.6% (Appendix A).

#### 2.5.2. Catalase (CAT)

CAT activity was determined using a UV-VIS spectrophotometer. This was expressed in sec^−1^ per gram of hemoglobin [23]. The coefficient of variation between replicas had to be less than or equal to 4.9% (Appendix A).

#### 2.5.3. Superoxide Dismutase (SOD)

SOD activity was measured using an SD125 Ransod kit (Randox Ltd. Crumlin, United Kingdom) [24]. The coefficient of variation between replicas had to be less than or equal to 5.1% (Appendix A).

#### 2.5.4. Glutathione Reduced (GSH) and Oxidized (GSSG)

GSH was analyzed by the glutathione-S-transferase assay described by Akerboom and Sies [25]. Glutathione oxidized form and glutathione disulfide (GSSG) were determined in a similar way to GSH as shown above, as described by Asensi [26]. The coefficient of variation for GSH between replicas must be less than or equal to 4.1% (Appendix A).

#### 2.5.5. Hemoxygenase 1 (HO1)

A commercial kit was used based on the Enzyme-Linked ImmunoSorbent Assay (ELISA) method (Shanghai BlueGeneBiotech Co., Ltd., Shanghai, China) with a detection limit of 0.1 ng/mL, according to the manufacturer’s instructions. The coefficient of variation between replicas must be less than or equal to 4.9% (Appendix A).

#### 2.5.6. Measurement of Cytokines IL6, TNFα and MCP1

These assays employed the quantitative sandwich enzyme immunoassay technique (DRG Instruments GmbH, Marburg, Germany), according to the manufacturer’s instructions. A monoclonal antibody specific for IL6, TNFα and MCP1 was precoated onto a microplate. Standards and samples were placed into the wells, and any IL6, TNFα and MCP1 present were bounded by the immobilized antibody. After washing away any unbounded substances, an enzyme-linked polyclonal antibody specific for IL6, TNFα and MCP1 was added to the wells. Following a wash to remove any unbounded antibody–enzyme reagent, a substrate solution was added to the wells, and color developed in proportion to the amount of IL6, TNFα and MCP1 bounded in the initial step. The color development was stopped, and the intensity of the color was measured. The coefficient of variation for IL6, TNFα and MCP1 between replicas must be less than or equal to 4.4, 6.4 and 4.7%, respectively (Appendix A).

#### 2.5.7. C reactive Protein (CRP)

For CRP-ultrasensitive (PCR-Turbilátex, Spinreact, Girona, Spain) detection, a turbidimetric test was used for the quantification of low serum CRP levels, according to the manufacturer’s instructions. Latex particles coated with anti-human CRP antibodies were agglutinated by CRP that was present in the subject’s sample. The agglutination process caused an absorbance change proportional to the CRP concentration of the sample, and by comparison with a CRP calibrator of known concentration, the CRP content in the analyzed sample was determined. The coefficient of variation between replicas had to be less than or equal to 4.7%.

### 2.6. Statistical Analyses

Data analysis was conducted using IBM Social Sciences software (SPSS, version 21.0, Chicago, IL, USA). Descriptive statistics are presented as mean and standard deviation (SD). Levene’s and Shapiro–Wilk tests were applied to check the homogeneity and normality of the data, respectively. A group × time × moment ANOVA was conducted to analyze within-group and between-group differences in all dependent variables and for every time-point of measurement (baseline, post-P_MAX_ and post-REC) and in both moments (pre-test and post-test). In addition, the area under the curve (AUC), resulting from the integration of the three time-points of measurement taken during the rectangular test, was calculated for each variable. The AUC was used to analyze pre-post differences both within groups and between groups. The within-group differences in the AUC were analyzed by repeated-measures *t*-test, and between-group comparisons in the AUC were conducted by applying an independent samples T-test. Cohen’s d effect size (ES) (95% confidence interval) was calculated for all comparisons. Threshold values for ES statistics were as follows: > 0.2 small, > 0.5 moderate, > 0.8 large [27]. Significant differences were considered for *p* ≤ 0.05.

## 3. Results

### 3.1. Biomarkers of Antioxidants and Oxidants Endogenous

Obtained values for CAT, SOD, GSSG, GSH, GSSG/GSH and HO1 during the rectangular test, pre- and post-intervention, are presented in Table 3. For each parameter, within group changes at each time point (baseline, Post-P_MAX_ and Post- REC) during supplementation were evaluated. A significant increase in SOD activity was found for the 2S-hesperidin group in post-P_MAX_ (15.5%) and post-REC (16.3%), while the placebo showed a significant increase in SOD at baseline (18.1%), intragroup pre-post-intervention (Figure 2). In addition, a similar increase in SOD in the AUC was found in 2S-hesperidin (14.1%) and placebo (11.9%) in the intragroup statistical analysis, without significant differences between groups (Figure 3).

Additionally, a trend towards a decrease with a moderate size effect in GSSG levels at post-P_MAX_ (−17.7%) was found in 2S-hesperidin in the post-intervention intragroup statistical analysis. In contrast, a significant decrease with a large size effect in GSSG was observed in the placebo at post-P_MAX_ (−15.1%) after the intervention (Figure 2). When comparing baseline post-intervention between groups, 2S-hesperidin had lower GSSG values (−20.1%) than the placebo (Figure 2). After the analysis of the AUC intragroup, there was a decrease in GSSG (−14.6%) only in 2S-hesperidin, without differences between groups (Figure 3).

For GSH, a decrease was reported at baseline (−9.4%) and post-P_MAX_ (−10.7%) in 2S-hesperdin after the intervention. On the other hand, a significant decline was found at baseline (−8.3%) in the placebo (Figure 2). Intragroup AUC analysis of GSH showed a decrease in 2S-hesperidin (−9.5%) and the placebo (5.5%), without differences between groups (Figure 3).

After the intragroup analysis, HO1 significantly increased at post-REC (19.7%) in the placebo, while there was a non-significant increase with a moderate size effect in HO1 at post-P_MAX_ (22.8%) in 2S-hesperidin (Figure 2). Intragroup AUC analysis showed an increase in HO1 in the placebo (20.9%) without any differences between groups (Figure 3).

When we analyzed the intragroup TBARS data, we found a trend towards an increase with a moderate size effect at baseline (9.4%) in 2S-hesperidin, without significant differences between groups.

When results for each parameter at each time point during supplementation were compared between groups, no significant changes were found in any antioxidant-oxidant parameter.

### 3.2. Inflammatory Biomarkers

Table 4 shows the obtained values for inflammatory biomarkers IL6, TNFα, MCP1 and CRP during the rectangular test, pre- and post-intervention. Within-group changes for each parameter and time point (baseline, Post-P_MAX_ and Post- REC) during supplementation have been evaluated. The placebo group showed a significant decrease in IL6 at Post-P_MAX_ (−35.7%), without significant changes in 2S-hesperidin (Figure 4), in the intra-group comparison pre- and post-intervention. However, after the intragroup analysis, we reported a decline in the AUC of IL6 (−33.0%) in the placebo after the supplementation period, without differences between groups (Figure 5).

Regarding TNFα, a significant drop in levels at baseline (−13.3%) and post-P_MAX_ (−14.5%) was found in the placebo (Figure 4). In addition, intragroup AUC analysis of TNFα found a decrease (−12.4%) in the placebo without differences between groups (Figure 5).

Significant decreases were observed in MCP1 at baseline (−20.2%) and post-REC (−26.1%) in 2S-hesperidin. In the placebo, significant decreases were also observed in MCP1 at baseline (−23.0%) and post-P_MAX_ (−14.2%) in the post-intervention intragroup statistical analysis (Figure 4). When comparing MCP1 at different times (baseline, Post-P_MAX_ and Post-REC) of the rectangular test post-intervention between groups, 2S-hesperidin had lower MCP1 values (−17.6%, −17.4% and −18.4%, respectively) than the placebo (Figure 4). In addition, a similar decrease in the AUC was found in 2S-hesperidin (−17.5%) and the placebo (16.1%) in intragroup statistical analysis, but in the case of 2S-hesperidin, a moderate size effect was observed, without significant differences between groups (Figure 5).

No significant within-group changes were reported for CRP in any group (Figure 4).

When results for each parameter at each time point during supplementation were compared between groups, no significant changes were found in any inflammatory parameter.

### 3.3. Hesperidin Metabolites Urine

Different hesperidin metabolites, mainly hesperetin glucuronides and sulfates, were analyzed in urine after Cardiose^®^ intake. The main metabolite detected was hesperetin-3-glucuronide, representing 78.9 ± 5.0% (*n* = 20) of the total, while hesperetin-7-glucuronide and hesperetin-7-sulfate made up 6.9 ± 2.9% (*n* = 20) and 14.7 ± 4.1% (*n* = 20) of the excreted metabolites. Despite the similarities in the excreted metabolites’ profiles, a large interindividual variability was observed in the number of excreted hesperidin metabolites ranging from 2.3 to 37.5 μmol.

## 4. Discussion

This study evaluated the effect of 8-week supplementation with 500 mg/d of 2S-hesperidin or placebo on antioxidant and inflammatory status in amateur cyclists during a rectangular cycle-ergometer test. To the best of our knowledge, this is the first study that examines the effect of chronic 2S-hesperidin intake on the antioxidant and inflammatory status of athletes at baseline, during and after exercise. In the rectangular test, oxidative status improved (↓GSSG AUC) after the 2S-hesperidin intervention, but not with the placebo. In addition, significant improvements in antioxidant capacity (↑SOD) after maximal exercise (Post-P_MAX_) and inflammatory status after the acute recovery phase (↓MCP1) were found in the 2S-hesperidin group compared to the placebo (baseline and Post-REC).

### 4.1. Changes in Endogenous Antioxidant Markers

SOD activity is usually increased after training, as an exercise-mediated adaptation [28]. In contrast, a previous study showed no increases in SOD activity in untrained individuals after an 8-week moderate training program (35-min aerobic cycle, 3 times/week) [29]. Conversely, we observed a maintenance of SOD from baseline to Post-P_MAX_ and an increase to Post-REC in the post-intervention rectangular test, with significant increases in Post-P_MAX_ and Post-REC pre-post-intervention in 2S-hesperidin. However, an increase in SOD activity levels, evaluated as the AUC, was observed for both groups during intervention. In amateur cyclists, the acute intake of 2S-hesperidin (single dose; 500 mg) led to no significant decrease in SOD at baseline [9]. In animals, 2S-hesperidin supplementation (200 mg/kg for three days per week), along with a 5-week training program, led to no significant changes in SOD activity in rats after an exhaustive exercise test [10]. Additionally, 2S-hesperidin significantly increased SOD activity in heart tissues, which was attenuated by doxorubicin (induced cardiac toxicity) treatment [30]. The 2S-hesperidin antioxidant capacity enhancement may be explained by the antioxidant characteristics of this molecule, related to hydroxyl groups in its B-ring [31]. In addition, Parhiz et al. found that hesperidin had significant radical scavenging activity and prevented H_2_O_2_-induced oxidative damage on the cellular membranes of red blood cells, with radical scavenging activities comparable to ascorbic acid and trolox (a vitamin E derivative) [32]. Furthermore, 2S-hesperidin shows a neutralizing effect on non-enzymatic lipid peroxidation and superoxide, hydroxyl, peroxynitrite, and nitric oxide radicals [4,31], leading to a lower depletion of antioxidant enzymes and allowing the maintenance of high antioxidant levels, even after exercise-induced oxidative stress.

Another mechanism that has been reported in vitro and in animal models, but has yet to be confirmed in humans, is the improvement of the antioxidant status through a nutrigenetic effect. Hesperidin has shown increased regulation of respiratory nuclear factor 2 (NRF2) [33]. NRF2 is a basic leucine zipper transcription factor that binds and activates the antioxidant response element in the promoters of many antioxidant and detoxification genes encoding proteins, such as SOD, glutathione, thioredoxin and HO1, and thus it promotes the regulation of the intracellular redox environment [34]. Interestingly, flavonoids have been proposed as inductors of the expression of genes related to enzymes of the endogenous antioxidant system through the activation of the NRF2 transcription pathway [35]. The higher SOD activity at the end of maximal effort and after a short recovery period after the intervention indicates that the chronic intake of 2S-hesperidin improves antioxidant capacity at maximal effort and in the acute phase of recovery in amateur cyclists.

The oxidation of GSH to GSSG is a sensitive marker of oxidative stress [36]. In addition, a GSH decrease and GSSG and GSSG/GSH ratio increases have been observed in professional cyclists after competitions [37]. When comparing both groups at baseline after the intervention, the 2S-hesperidin group had lower GSSG values than the placebo, indicating lower levels of oxidative stress. This is in line with the decrease found in the AUC (GSSG) in 2S-hesperidin, indicating a decrease in oxidative stress when considering the whole rectangular test, which may be related to detraining adaptation. In fact, lower training volumes and intensities are associated with lower levels of GSH and GSSG in professional cyclists [38], which was also found Post-P_MAX_ in the placebo. Therefore, this decrease in both groups is due to a lower exposure to high levels of free radicals leading to a maladaptation in the glutathione antioxidant system. The main advantage of incorporating the AUC in this study is that it allows us to precisely define the duration and magnitude of the variable being evaluated, which cannot be done in a point-by-point comparison [39]. Despite the fact that there are no previous studies in humans evaluating the effects of chronic hesperidin intake on GSH and GSSG, instead, non-significant decreases in GSH, GSSH and the GSSG/GSH ratio were observed after a repeated sprint test in amateur cyclists after a single-dose of 2S-hesperidin (500 mg) [9]. In the same way, pathological animal models have shown the positive effect of hesperidin supplementation on these glutathione markers (↑GSH and ↓GSSG) [40,41].

It is known that regulation exists between GSH and GSSG by the enzymes glutathione reductase (GR) and GPx to maintain a balance between both molecules and avoid an increase in ROS [42]. The changes observed in GSSG in the experimental group could be due to the modulation of GPx and GR activity, which was not measured in this study. In addition, another factor that may influence the GSSG/GSH ratio is the levels of nicotinamide adenine dinucleotide phosphate (NADPH), which are used with an indispensable cofactor by the GR and GPx enzymes to synthesize the GSH and GSSG forms [43]. In this context, NADPH donates two electrons to reduce GSSG to GSH by GR; the recycled GSH can then be used to reduce H_2_O_2_ to water by GPxs [44]. In addition, increased glucose-6-phosphatase dehydrogenase (G6PD) (a major source of cytosolic NADPH) activity by genetic or pharmacological means has been seen to raise cellular stores of NAPDH and GSH, promote the detoxification of ROS, and increase cell viability in primary vascular endothelial and smooth muscle cells in vitro [45]. Increased G6PD activity is positively correlated with increased GR activity, where hesperidin was capable of restoring the activity of G6PD in rats [46]. In addition, Salvemini et al. [47] reported that a three-fold increase in G6PD activity resulted in a two-fold increase in GSH levels, as well as a very significant increase in resistance to oxidative stress. As we can see, the GSSG/GSH ratio can be modulated by different components involved in the endogenous antioxidant system, which makes it difficult to explain its changes. Therefore, the chronic intake of 2S-hesperidin could decrease GSSG levels (evidenced by ↓ AUC), indicating a better antioxidant state in the rectangular test, but specifically immediately after exercise. This would facilitate faster post-training recovery or competition for cyclists.

In relation to HO1, in the placebo there was an increase Post-REC with an increasing trend in baseline and Post-P_MAX_ post-intervention; however, after 2S-hesperidin supplementation, no significant change was seen, but there was a moderate effect in Post-P_MAX_. The high variability in the HO1 data in 2S-hesperidin may have been the consequence of no significant changes being observed. What is clear is that amateur-level cycling for 8 weeks improves HO1 levels.

Although there is no clear pattern of improvement in antioxidant markers in 2S-hesperidin, there is an improvement in certain components (↑SOD, ↓GSSG) of the endogenous antioxidant system measured in this study and at key times during recovery. However, further studies are needed to provide clarity on this issue.

### 4.2. Changes in Inflammatory Markers

The production of ROS at the mitochondria of the working muscle stimulates the production of myokines or pro-inflammatory cytokines [48]. IL6 (inflammatory cytokine) plasma levels can increase up to 100-fold after exercise, and circulating muscle-derived IL6 levels are closely related to the duration and intensity of exercise [49]. To our knowledge, no studies have evaluated the effects of 2S-hesperidin intake on inflammatory markers in humans. In this study, IL6 levels increased during the first and second rectangular test from baseline to Post-P_MAX_ in both groups, but there were different trends from Post-P_MAX_ to Post-REC in the second rectangular test (↓2S-hesperidin and ↑placebo). A significant decrease in IL6 during the recovery stage was observed in the placebo, post-intervention. Other flavonoids, such as cocoa-derived flavanols, have also failed to inhibit the increase in IL6 after intense exercise (75% of peak power output for 30 min) in cyclists [50]. We believe that the high variability in the IL6 data was a factor that did not allow us to find significant intra- and intergroup differences. In addition, IL6 values in the placebo were quantitatively higher than those of 2S-hesperidin, which may favor a significant decrease in IL6 after the reduction in the training load performed by cyclists from post-season to pre-season, as was the period in which the study was conducted (from the end of September to the end of December). As IL6 is known to stimulate the expression of TNFα [51], a decrease in IL6 levels in the placebo would lead to a decrease in TNFα levels (baseline and Post-P_MAX_). Since there is less training load, there is less induction of oxidative stress [38] and, consequently, less stimulation of the inflammatory system.

However, numerous in vitro studies (inflammatory models) have shown the ability of hesperidin to lower IL6 levels and TNFα [52,53,54,55]. A recent study in trained animals showed that hesperidin intake (200 mg/kg for three days per week) during 5 weeks prevented an increase in IL6 levels in peritoneal macrophages after an exhausting exercise [56]. Interestingly, in this study, a significant increase in IL6 after an exhausting exercise, from pre-training to post-intervention, was observed in the placebo group. Hesperidin intake has also led to a decrease in IL6 in a rheumatoid arthritis rat model [57]. In rats, the intake of alcohol to induce a gastric ulcer increased the expression of cyclooxygenase-2 mRNA and decreased GPx, SOD, and CAT, but the intake of hesperidin reversed these changes, improving the antioxidant and inflammatory status [58]. In addition, in a model of Alzheimer’s disease in mice, treatment with hesperidin (40 mg/kg, 90 days intragastric) increased HO1 and decreased levels of TNFα, CRP, NF-κB and MCP1, suppressing oxidative stress and inflammation [59].

A hypothesis has been generated at the nutrigenomic level of how the intake of hesperidin can improve the inflammatory state, related to the activation of the Akt/NRF2 axis and the inhibition of NF-κB [59], with the latter being a transcription factor well known for its role in the innate immune response and a transcriptional activator of inflammatory mediators such as cytokines [60]. On the other hand, NRF2 is not only important for redox signaling, but also for the attenuation of the inflammatory mediator synthesis [59]. In this sense, the impairment of NRF2 signaling by ultraviolet B (UVB) was reversed by the topical application of hesperidin methyl chalcone, which inhibited the production of the cytokines TNFα, IL-1β, IL6, and IL-10 that had been induced by UVB irradiation in hairless mice [61]. This suggests that there is a connection between the antioxidant and inflammatory status and their signaling pathways. In our case, the group ingesting 2S-hesperidin did not experience a significant decrease in TNFα.

MCP1 is another inflammatory cytokine that increases after exercise in plasma [62]. In our study, lower MPC-1 levels during the whole exercise (AUC) were observed after supplementation in both groups. This decrease was statically significant at baseline and during the recovery phase for the 2S-hesperidin supplemented group. In addition, when comparing between groups at different post-intervention test times, the 2S-hesperidin group had lower levels compared to the placebo. In previous studies with an acute lung damage model, both in vitro and in vivo, hesperidin has shown immunomodulatory effects, down-regulating the expression of MCP1 as well as other pro-inflammatory cytokines, such as IL6 and TNFα [52,63]. Precisely, treatment with hesperetin-7-O-glucuronide (5 mg kg^−1^) has been observed to decrease the MCP1 mRNA expression in rat aortic endothelial cells [63]. On the other hand, the oral administration of 100 or 200 mg/kg of hesperidin three times a week for four weeks in rats produced a decrease in the pro-inflammatory cytokines interferon-gamma-γ and MCP1 in the lymphocyte of the mesenteric lymph node [10]. Additionally, polyphenols and hesperidin can modulate gut microbial composition or functionality, which modulate the release of microbial-derived metabolites [64]. In addition, hesperidin has the ability to inhibit the growth of harmful bacteria, such as Escherichia coli, Pseudomonas aeruginosa, Prevotella spp., Porphyromonas gingivalis and Fusobacterium nucleatum, among others [3]. In particular, hesperidin can increase the abundance of Faecalibacterium prausnitzii, which inhibits NF-κB activation and consequently attenuates the inflammatory response [65]. The inhibitory capacity of hesperidin in some bacteria may modify the composition of the intestinal microbiota acting as an immunomodulator and anti-inflammatory (↓IL-1β, TNFα, and IL6), with a direct relationship between the two effects [3]. In contrast, the effects of quercetin (flavonoid) intake (1g/day) for 3 weeks in trained cyclists were evaluated by Nieman et al. [62]. In this study, no significant changes in MCP1 plasma levels were observed after a 3-week supplementation and a 3-day period in which subjects cycled for 3 h/day at ~57% maximal work rate. Muscle biopsies showed a within-group significant post-exercise increase in muscle cytokine mRNA expression for IL6 and TNFα, but without differences between the quercetin and placebo groups [62]. No anti-inflammatory effect was observed after the intake of quercetin. On the other hand, the placebo group showed a decrease in MCP1 at baseline and Post-P_MAX_ at post-intervention, possibly related to the decrease in TNF-α, as a positive correlation between MCP-1 and TNF-α concentrations after short-term exercise training has been previously demonstrated [66], indicating a relationship between these two cytokines.

Although there is no clear pattern of improvement in inflammatory markers in 2S-hesperidin, there is an enhancement in MCP1 (baseline and Post-REC) compared to the placebo in the second rectangular test at all points and at key times during recovery. However, further studies are needed to bring clarity to this question.

These 2S-hesperidin properties such as antioxidant and anti-inflammatory properties may be related: a decrease in oxidative stress during the exercise maximum intensity could modulate the inflammatory state in the acute phase of recovery. As has been shown in the studies presented in this publication, there is a close relationship between antioxidant and inflammatory status and their signaling pathways. Redox balance can be altered during periods of high intensity physical exercise and low rest periods, leading to a chronic oxidative stress state [15]. Moreover, high oxidative stress levels can inhibit exercise physiological adaptations, reducing performance and leading to overtraining [15]. Therefore, an optimal redox homeostasis is essential for a proper muscle physiological function (i.e., antioxidant status, biochemistry, signaling, bioenergetics and muscle contraction).

The effects of antioxidant supplementation on performance are a controversial topic, which still needs additional research. On one hand, it has been pointed out that the use of antioxidant substances may help to maintain optimal ROS levels in the muscle, avoiding possible decreases in performance [15]. On the other hand, it has been hypothesized that chronic antioxidant intake can hinder training adaptations, negatively affecting performance [17]. Different studies show that antioxidant intake does not prevent the exercise-induced activation of redox-sensitive signaling pathways [67]. A recent publication summarized the performance measurements that were carried out in this same intervention trial, along with the antioxidant and inflammatory marker results reported in this paper [7]. In this trial, amateur cyclists’ supplementation (8 weeks) with 2S-hesperidin (500 mg/day) led to an increase in power production at estimated functional threshold power (2.3% = 6.40 W; *p* = 0.049) and maximum power (1.9% = 7.40 W; *p* = 0.049) during an incremental test after the intervention [7]. Thus, 2S-hesperidin does not appear to interfere with training-induced adaptations, improving performance while avoiding oxidative stress and inflammation.

The study described in this paper has some limitations. One limitation was the short recovery time after the rectangular test (20 min after exercise), in which changes in antioxidant and inflammatory markers were evaluated. Measurements 24 and 48 h after exercise would have provided valuable additional information; however, funding constraints made it impossible. Additionally, a larger sample would have given more statistical power to the reported results due to the high individual variability in some markers. Given the few studies carried out in this field, future research could shed light on the effectiveness of 2S-hesperidin as an ergogenic aid with antioxidant and inflammatory effects.

Differences with current results may be related to the different stage of the season in which studies were done in the sample used, and the different aerobic and anaerobic demand profiles of the used tests. In the same way, one of the factors influencing the variability of 2S-hesperidin effects in different studies may be its pharmacokinetics, and the resulting exposure of the body to hesperidin metabolites. It has been described that the concentration of 2S-hesperidin metabolites in plasma reaches its maximum peak 5–7 h after intake, being almost completely eliminated after 24 h. In the urine, the maximum peak of metabolites is usually found at 24 h of 2S-hesperidin intake, and its total excretion occurs after 48 h [68].

## 5. Conclusions

Supplementation with 2S-hesperidin (500mg/d) for 8 weeks improves the post-rectangular test antioxidant (↑SOD and ↓AUC-GSSG) and inflammatory status during the acute phase of post-exercise recovery (↓MCP1). This modulation in antioxidant and inflammatory markers can help cyclists to improve their recovery after intense efforts or long exercise sessions that, due to their characteristics, led to an increase in inflammation and oxidative stress. Unlike other polyphenols, 2S-hesperidin supplementation does not appear to interrupt adaptations produced by training in amateur cyclists, enhancing their performance [7].

## Figures and Tables

**Figure 1 antioxidants-10-00432-f001:**
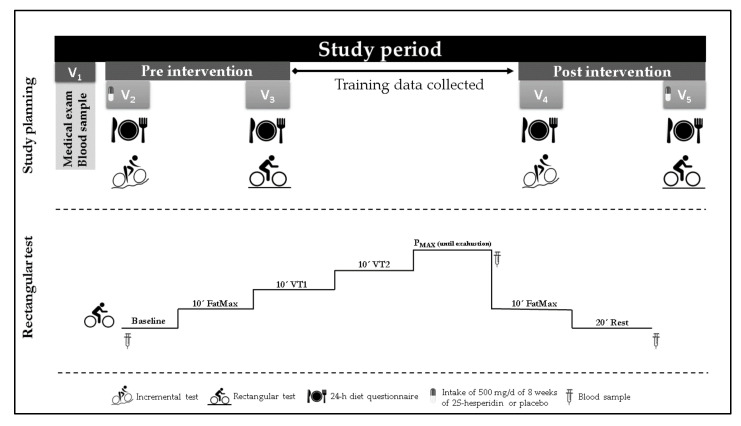
Study planning and rectangular test protocol.

**Figure 2 antioxidants-10-00432-f002:**
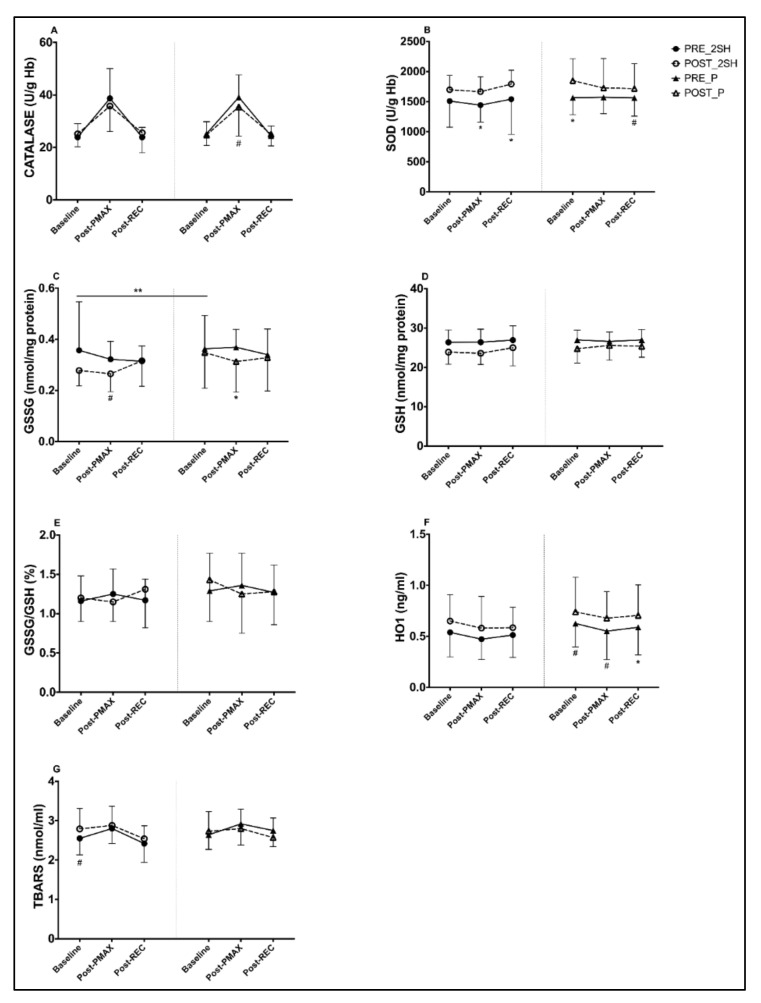
Differences between pre- and post-intervention intragroup in antioxidant and oxidant parameters at different points of the rectangular test (**A**–**G**). (**C**), a significant difference (*p* = 0.04) appears, comparing baseline of the second rectangular test between groups. * *p* < 0.05. # *p* = 0.05–0.06. ** *p* < 0.05 between post-intervention time points of rectangular test between groups (2S-hesperidin vs. placebo).

**Figure 3 antioxidants-10-00432-f003:**
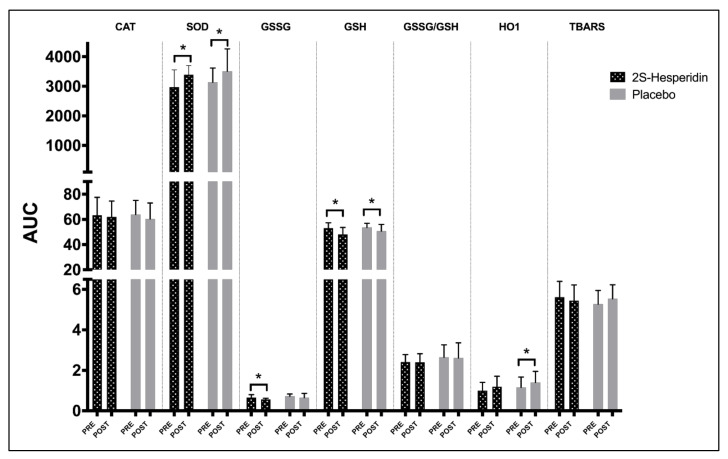
Intragroup differences between AUC (pre- and post-intervention) in antioxidant parameters. * *p* < 0.05. There were no significant differences between groups in AUC.

**Figure 4 antioxidants-10-00432-f004:**
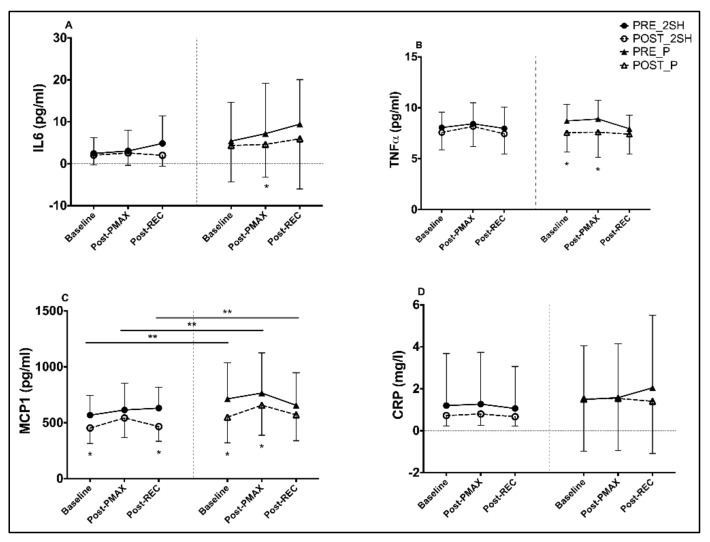
Differences between pre- and post-intervention intragroup and intergroup in inflammatory parameters at different points of the rectangular test (**A**–**D**). (**C**) a significant difference appears, comparing baseline (*p* = 0.043), Post-P_MAX_ (*p* = 0.026) and Post-REC (*p* = 0.045) of the second rectangular test between groups. * *p* < 0.05. ** *p* < 0.05 between post-intervention time points of rectangular test between groups (2S-hesperidin vs. placebo).

**Figure 5 antioxidants-10-00432-f005:**
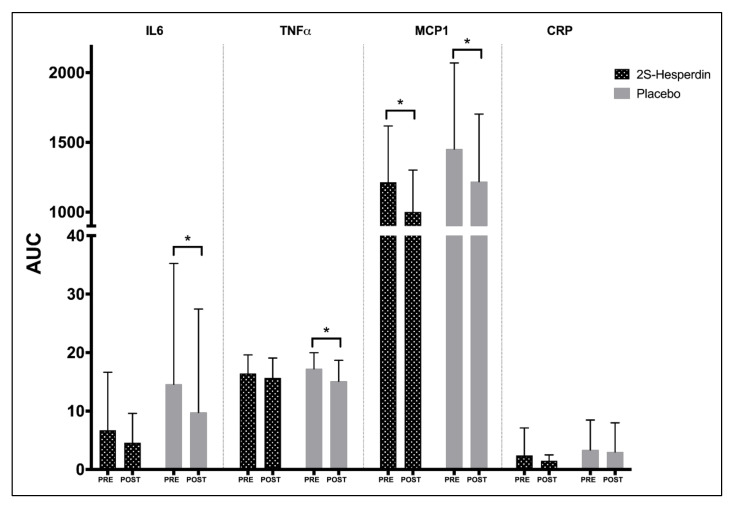
Intragroup differences between AUC (pre- and post-intervention) in inflammatory parameters. * *p* < 0.05. There were no significant differences between groups in AUC.

**Table 1 antioxidants-10-00432-t001:** Baseline general characteristics and training variables of the cyclists.

	2S-Hesperidin	Placebo	*p*-Value
**Age (years)**	35.0 (9.20)	32.6 (8.90)	0.407
**Body mass (kg)**	71.0 (6.98)	70.4 (6.06)	0.773
**Height (cm)**	175.3 (6.20)	176.5 (6.10)	0.541
**BMI (kg·m^−2^)**	23.1 (1.53)	22.6 (1.43)	0.292
**BF (%)**	8.9 (1.63)	9.0 (1.64)	0.803
**VO_2MAX_ (L·min^−1^)**	3.99 (0.36)	3.98 (0.63)	0.971
**VO_2MAX_ (mL·kg^−1^·min^−1^)**	57.5 (6.97)	57.9 (9.53)	0.88
**HR_MAX_ (bpm)**	184.9 (11.11)	183.2 (8.68)	0.593
**VT1 (%)**	50.9 (5.63)	50.0 (4.78)	0.61
**VT2 (%)**	84.9 (5.85)	84.1 (5.70)	0.644
**Training variables**	**2S-Hesperidin**	**Placebo**	***p*-value**
**Total distance (km)**	1121.12 (534.99)	1082.43 (810.46)	0.868
**HR_AVG_ (bpm)**	144.76 (8.88)	137.48 (13.11)	0.067
**W_AVG_ (W)**	174.86 (15.79)	163.47 (32.49)	0.435
**RPE**	6.34 (0.82)	6.33 (1.16)	0.975

Values are expressed as mean (SD). BMI = body mass index; BF = body fat; VO_2max_ = maximum oxygen volume; VT1 = ventilatory threshold 1 (aerobic); VT2 = ventilatory threshold 2 (anaerobic); Total distance = of all the training sessions carried out during the study period; HRavg = average heart rate of all the training sessions carried out during the study period; Wavg = average power output of all training sessions during the study period and RPE = rating of perceived exertion of all training sessions during the study.

**Table 2 antioxidants-10-00432-t002:** Between-group comparisons of dietary intake of cyclists.

Pre-Intervention				Post-Intervention		
	2S-Hesperidin	Placebo	*p*-Value	2S-Hesperidin	Placebo	*p*-Value
**Kcal**	2163.6 (519.02)	2100.2 (515.77)	0.708	1974.1 (377.97)	2133.5 (437.98)	0.237
**Kcal/BM**	31.1 (9.34)	30.2 (8.71)	0.768	27.9 (6.53)	30.3 (6.46)	0.249
**CHO (g)**	245.7 (73.46)	222.0 (69.68)	0.312	216.6 (63.47)	248.3 (58.15)	0.117
**CHO/BM**	3.5 (1.31)	3.2 (1.14)	0.416	3.1 (1.08)	3.5 (0.94)	0.173
**PRO (g)**	113.5 (25.21)	115.2 (25.37)	0.837	109.0 (23.05)	101.5 (23.67)	0.332
**PRO/BM**	1.6 (0.41)	1.7 (0.48)	0.778	1.5 (0.35)	1.5 (0.42)	0.596
**LP (g)**	80.8 (27.24)	83.5 (23.65)	0.739	71.5 (17.61)	71.6 (18.89)	0.985
**LP/BM**	1.2 (0.45)	1.2 (0.37)	0.758	1.0 (0.27)	1.0 (0.29)	0.823

Values are expressed as mean (SD). Kcal = kilocalories; CHO = carbohydrates; PRO = protein; LP = lipids; BM = body mass. The mean values correspond to the average of all 24-h diet recall data collected at pre-intervention (visits 2, 3 and 4) and post-intervention (visits 5, 6 and 7).

**Table 3 antioxidants-10-00432-t003:** Changes in enzymes and peptides endogenous antioxidants before, during and after rectangular test comparing pre- and post-intervention.

		2S-HESPERIDIN				PLACEBO				Between-Group Comparison			
		Baseline	Post- P_MAX_	Post- REC	AUC	Baseline	Post- P_MAX_	Post- REC	AUC	ΔBaseline	ΔPost-P_MAX_	ΔPost-REC	ΔAUC
**CAT**	Pre-Int	23.89	38.66	23.89	63.29	25.15	39.14	24.34	63.88				
		(5.14)	(11.4)	(3.75)	(14.2)	(4.54)	(8.51)	(3.85)	(11.20)				
	Post-Int	25.17	35.57	25.5	61.91	24.78	35.45	25.01	60.34				
		(4.95)	(9.66)	(7.65)	(12.62)	(4.02)	(11.12)	(4.41)	(12.74)				
	*p-value*	0.272	0.148	0.263	0.526	0.757	0.058	0.633	0.188	0.316	0.76	0.636	0.527
	*Effect size*	0.24	0.26	0.41	0.09	0.08	0.42	0.17	0.30	0.32	0.07	0.15	0.2
**SOD**	Pre-Int	1509	1442	1541	2971	1566	1573	1563	3138				
		(435.05)	(282.29)	(280.16)	(584.34)	(284.36)	(274.41)	(303.05)	(479.34)				
	Post-Int	1698	1666	1792	3391	1849	1727	1718	3511				
		(238.55)	(246.110	(231.2)	(308.32)	(364.77)	(491.32)	(412.24)	(754.07)				
	*p*-value	0.11	**0.045**	**0.004**	**0.011**	**0.009**	0.124	0.057	**0.023**	0.449	0.699	0.402	0.826
	*Effect size*	0.42	0.76	0.86	0.69	0.95	0.54	0.49	0.75	0.2	0.16	0.27	0.07
**GSSG**	Pre-Int	0.357	0.322	0.314	0.657	0.363	0.369	0.34	0.720				
		(0.19)	(0.07)	(0.060	(0.14)	(0.13)	(0.07)	(0.1)	(0.11)				
	Post-Int	0.278	0.265	0.316	0.561	0.348	0.313	0.328	0.650				
		(0.06)	(0.07)	(0.1)	(0.06)	(0.14)	(0.12)	(0.13)	(0.21)				
	*p*-value	0.08	**0.058**	0.963	**0.016**	0.734	**0.049**	0.719	0.219	0.326	0.979	0.772	0.627
	*Effect size*	0.39	0.75	0.02	0.64	0.11	0.82	0.11	0.59	0.31	0.01	0.09	0.12
**GSH**	Pre-Int	26.37	26.41	26.93	53.07	26.99	26.61	26.98	53.60				
		(3.14)	(3.31)	(3.66)	(4.25)	(2.46)	(2.40)	(2.66)	(3.30)				
	Post-Int	23.88	23.59	25.00	48.03	24.76	25.58	25.38	50.65				
		(3.09)	(2.86)	(4.61)	(5.57)	(3.66)	(3.78)	(2.80)	(5.30)				
	*p*-value	**0.014**	**0.029**	0.081	**0.011**	**0.042**	0.335	0.110	**0.027**	0.862	0.246	0.812	0.343
	*Effect size*	0.76	0.82	0.51	1.14	0.87	0.41	0.58	0.86	0.06	0.37	0.08	0.3
**GSSG/** **GSH**	Pre-Int	1.16	1.25	1.17	2.41	1.29	1.36	1.27	2.65				
		(0.32)	(0.32)	(0.27)	(0.37)	(0.48)	(0.41)	(0.35)	(0.61)				
	Post-Int	1.20	1.15	1.31	2.40	1.43	1.25	1.28	2.61				
		(0.30)	(0.25)	(0.49)	(0.42)	(0.53)	(0.50)	(0.42)	(0.75)				
	*p*-value	0.770	0.390	0.272	0.950	0.338	0.361	0.956	0.082	0.608	0.938	0.431	0.903
	*Effect size*	0.11	0.29	0.52	0.02	0.27	0.26	0.02	0.06	0.16	0.02	0.25	0.04
**HO1**	Pre-Int	0.539	0.473	0.513	0.998	0.626	0.551	0.589	1.158				
		(0.24)	(0.20)	(0.22)	(0.41)	(0.23)	(0.28)	(0.27)	(0.51)				
	Post-Int	0.650	0.581	0.585	1.198	0.740	0.678	0.705	1.400				
		(0.26)	(0.31)	(0.20)	(0.51)	(0.34)	(0.26)	(0.30)	(0.55)				
	*P*-value	0.135	0.172	0.281	0.081	0.059	0.066	**0.027**	**0.038**	0.972	0.849	0.548	0.789
	*Effect size*	0.44	0.53	0.32	0.47	0.47	0.43	0.41	0.46	0.01	0.06	0.19	0.09
**TBARS**	Pre-Int	2.55	2.80	2.41	5.27	2.64	2.92	2.75	5.62				
		(0.42)	(0.38)	(0.48)	(0.68)	(0.37)	(0.54)	(0.41)	(0.78)				
	Post-Int	2.79	2.88	2.54	5.54	2.73	2.80	2.57	5.45				
		(0.52)	(0.49)	(0.33)	(0.69)	(0.50)	(0.49)	(0.50)	(0.77)				
	*P*-value	**0.052**	0.519	0.393	0.246	0.527	0.493	0.216	0.551	0.399	0.406	0.134	0.227
	*Effect size*	0.55	0.22	0.25	0.38	0.22	0.21	0.42	0.21	0.27	0.27	0.48	0.39

Values are expressed as mean (SD). Abbreviations: AUC = area under curve; CAT = catalase; SOD = superoxide dismutase; GSH = reduced glutathione; GSSG = oxidized glutathione; % GSSG/GSH = oxidized/reduced glutathione ratio; HO1 = hemoxygenase 1; TBARS = thiobarbituric acid reactive substances and SD = standard deviation. Group comparison = *p*-value comparison of Δ pre-post intervention between groups at different times (baseline, post-P_MAX_ and post-REC) of rectangular test.

**Table 4 antioxidants-10-00432-t004:** Changes in inflammatory status markers before, during and after rectangular test comparing pre- and post-intervention.

		2S-Hesperidin				Placebo				Between-Group Comparison			
		Pre	Post- P_MAX_	Post- REC	AUC	Pre	Post- P_MAX_	Post- REC	AUC	ΔBaseline	ΔPost-P_MAX_	ΔPost-REC	ΔAUC
**IL6**	Pre-Int	2.46	3.05	4.85	6.71	5.41	7.17	9.44	14.59				
		(3.78)	(4.92)	(6.55)	(9.94)	(9.26)	(12.02)	(10.62)	(20.67)				
	Post-Int	2.04	2.57	2.01	4.59	4.35	4.61	5.96	9.77				
		(2.32)	(2.94)	(2.61)	(5.01)	(8.65)	(7.78)	(11.98)	(17.69)				
	*p*-value	0.537	0.695	0.128	0.255	0.129	**0.045**	0.065	**0.021**	0.514	0.243	0.807	0.31
	Effect size	0.11	0.10	0.42	0.20	0.11	0.20	0.31	0.22	0.21	0.39	0.08	0.33
**TNFα**	Pre-Int	8.06	8.43	7.97	16.44	8.71	8.90	7.94	17.22				
		(1.51)	(2.07)	(2.08)	(3.17)	(1.62)	(1.83)	(1.35)	(2.76)				
	Post-Int	7.58	8.17	7.44	15.68	7.55	7.61	7.41	15.09				
		(1.70)	(1.98)	(2.00)	(3.40)	(1.89)	(2.46)	(1.94)	(3.6)				
	*P*-value	0.271	0.466	0.148	0.338	**0.015**	**0.038**	0.127	**0.021**	0.252	0.21	0.999	0.239
	Effect size	0.31	0.12	0.25	0.23	0.69	0.67	0.38	0.74	0.37	0.4	0	0.38
**MCP1**	Pre-Int	568	614	631	1214	714	766	656	1452				
		(177.05)	(240.06)	(184.77)	(403.32)	(324.12)	(359.13)	(291.75)	(617.47)				
	Post-Int	453	543	466	1002	550	657	571	1218				
		(137.74)	(174.45)	(131.81)	(299.31)	(228.61)	(268.48)	(231.280	(484.67)				
	*P*-value	**0.007**	0.096	**0.004**	**0.002**	**<0.001**	**0.026**	0.108	**0.021**	0.388	0.567	0.275	0.845
	Effect size	0.63	0.29	0.86	0.5	0.49	0.29	0.28	0.36	0.28	0.18	0.35	0.06
**CRP**	Pre-Int	1.200 (2.48)	1.268 (2.47)	1.063	2.399	1.487	1.579	2.049	3.347				
				(2.00)	(4.71)	(2.56)	(2.57)	(3.45)	(5.12)				
	Post-Int	0.724	0.798	0.669	1.494	1.517	1.539	1.404	3.000				
		−0.49	−0.54	(0.45)	(1.01)	(2.50)	(2.49)	(2.49)	(4.98)				
	*p*-value	0.521	0.523	0.634	0.398	0.981	0.943	0.402	0.832	0.606	0.658	0.819	0.774
	Effect size	0.18	0.18	0.19	0.18	0.01	0.01	0.18	0.07	0.16	0.14	0.07	0.09

Values are expressed as mean (SD). Abbreviations: AUC = area under curve; IL-6 = interleukin 6; TNFα = tumor necrosis factor α; MCP-1 = monocytes chemoattractant protein 1 and CRP = C-reactive protein. Group comparison = p-value comparison of Δ pre-post intervention between groups at different times (baseline, post-P_MAX_ and post-REC) of rectangular test.

## Data Availability

All data is contained within the article.

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
