# Peer review of "8-Week Supplementation of 2S-Hesperidin Modulates Antioxidant and Inflammatory Status after Exercise until Exhaustion in Amateur Cyclists"

_antioxidants, 2021, doi:10.3390/antiox10030432_

Round 1

Reviewer 1 Report

The researchers have evaluated over eight-week period whether daily intake of 2S-hesperidin (500mg/d) had the ability to modulate antioxidant-oxidant and inflammatory status in amateur cyclists. They report that supplementation with 2S-hesperidin improved antioxidant and inflammatory status during the acute phase of post-exercise recovery but did not interrupt adaptations produced by training.

This is a thorough and well-documented study. However, it addresses a specific question with respect to cyclists and issues with respect to fatigue/ adaptation to exercise. How could these findings be extrapolated to the wider population? Also, while increased ROS and inflammation are products of intense exercise and treatments that prevent them can be beneficial, these host responses are also cues for the body to stop or limit the exercise activity to prevent irreversible damage. I think that the authors need to emphasise although 2S-hesperidin can have beneficial effects on fatigue/adaptation to exercise in cyclists the implications of prolonged or long-term usage are unknown.

Ln 39                    ‘Hesperidin may be foun in two isomeric forms, 2S- and 2R-, being the 2S isomer the predominant in nature’           ‘Hesperidin may be found in two isomeric forms, 2S- and 2R-, the 2S isomer being predominant in nature’?    

Ln 60-62              Sentence is unclear to a non-specialist.

Ln 76-87              ‘it is hypothesized that rapid increases in ROS during intensive exercise may be a contributor to fatigue’. This whole paragraph appears without any preamble. What is the problem that is being addressed and why is it important? This information would allow the general readers the context for the study and its possible wider implications.

Ln 175                  by a nurse?

Ln 183                  blood?

Ln 187-189          Urine samples. Were these collected over the 24h before the visit or 24 hours afterwards or both?

Ln 287-290          Sentences are unclear.         

Figure 2 and others         ‘# = 0.05>0.06’. What does this mean? Is it p=0.05-0.06?

Ln 365                  typo.

Ln 410                  de-training adaptation?

Ln 476-477          split sentence

Ln 523-525          Either the ‘Since’ is superfluous or part of the sentence is missing.

Ln 571                  I would advise including a sentence at the end to emphasise that the long-term effects of the intake of 2S-hesperidin for this purpose remain unknown.              

Author Response

Reviewer 1

The researchers have evaluated over eight-week period whether daily intake of 2S-hesperidin (500mg/d) had the ability to modulate antioxidant-oxidant and inflammatory status in amateur cyclists. They report that supplementation with 2S-hesperidin improved antioxidant and inflammatory status during the acute phase of post-exercise recovery but did not interrupt adaptations produced by training.

We thank the reviewer for their constructive and helpful feedback on our manuscript. We have replied to each specific comment in the section below and have introduced the corresponding edits into the manuscript using Word’s track changes.

This is a thorough and well-documented study. However, it addresses a specific question with respect to cyclists and issues with respect to fatigue/ adaptation to exercise. How could these findings be extrapolated to the wider population? Also, while increased ROS and inflammation are products of intense exercise and treatments that prevent them can be beneficial, these host responses are also cues for the body to stop or limit the exercise activity to prevent irreversible damage. I think that the authors need to emphasise although 2S-hesperidin can have beneficial effects on fatigue/adaptation to exercise in cyclists the implications of prolonged or long-term usage are unknown.

Response: When extrapolating the results obtained in our study to the general population, we can say that in people who are new to physical exercise, where there greater oxidative stress may occur after exercise, the intake of 2S-Hesperidin could improve post-exercise recovery by reducing oxidative stress and consequently the inflammatory state.

Response: The aim of reducing oxidative stress and, consequently, decreasedtheinflammatory status, is due to the fact that cyclists have high levels of physical intensity in training combined with the high number of competitions (especially stage races (from 3 to 21 days of competition in a row) and they generate high levels of oxidative stress and modulating the inflammatory system. A supplement, such as 2S-Hesperidin, could help to improve post-competition recovery, which is very important in stage races (Tour de France, Vuelta de España, etc.). In addition, current data indicate that reactive oxygen and nitrogen species (ROS) and inflammatory pathways are the most likely mechanisms contributing to overtraining syndrome in skeletal muscle (DOI: 10.1016/j.redox.2020.101480). Therefore, treatments or supplements that can counteract the generated high oxidative stress over time could prevent or delay the onset of overtraining syndrome in athletes. 

Response: Following your suggestion, we have included the phrase "However, the implications of long-term or prolonged use are unknown" in lines 98-99.

Ln 39                    ‘Hesperidin may be foun in two isomeric forms, 2S- and 2R-, being the 2S isomer the predominant in nature’           ‘Hesperidin may be found in two isomeric forms, 2S- and 2R-, the 2S isomer being predominant in nature’?    

Response: Amended

Ln 60-62              Sentence is unclear to a non-specialist.

Response: Following your suggestion, we have modified the sentence in line 62.

Ln 76-87              ‘it is hypothesized that rapid increases in ROS during intensive exercise may be a contributor to fatigue’. This whole paragraph appears without any preamble. What is the problem that is being addressed and why is it important? This information would allow the general readers the context for the study and its possible wider implications.

Response: Following your suggestion, we have inserted a preamble to the sentence you indicate. Lines 77-79.

Ln 175                  by a nurse?

Response: Amended. Line 177.

Ln 183                  blood?

Response: Amended. Line 199.

Ln 187-189          Urine samples. Were these collected over the 24h before the visit or 24 hours afterwards or both?

Response: Amended. Line 203.

Ln 287-290          Sentences are unclear.         

Response: Following your suggestion, we have modified these sentences. Lines 290-291.

Figure 2 and others         ‘# = 0.05>0.06’. What does this mean? Is it p=0.05-0.06?

Response: Amended

Ln 365                  typo.

Response: Amended. Line 360. We think that you are referring to the type of exercise or timing of the rectangular test. If we have not answered your suggestion, please let us know.

Ln 410                  de-training adaptation?

Response: Amended. Line 442.

Ln 476-477          split sentence

Response: Amended. Lines 515-517.

Ln 523-525          Either the ‘Since’ is superfluous or part of the sentence is missing.

Response: Amended. Line 571.

Ln 571                  I would advise including a sentence at the end to emphasise that the long-term effects of the intake of 2S-hesperidin for this purpose remain unknown.   

Response: We understand your concern, but to include that sentence would be to negate the findings of the study we published in December, part of the same project as the current study, in which we found improvements in performance after 8 weeks of 2S-hesperidin in amateur cyclists (doi.org/10.3390/nu12123911).

Author comment: We appreciate all the comments made on our manuscript, which helped improve it’s quality.

Reviewer 2 Report

Following 8-week supplementation with 2S-hesperidin, the authors measured some antioxidative and inflammatory parameters in amateur cyclists after exercise until exhaustion. In my opinion, the manuscript must be improved before considering publication.

  1. The manuscript is very hard to follow. English must be improved (professional assistance is suggested). The design of the study, as well as the statistical analysis, are not described in an understandable manner for an average reader. There are many groups and time points, and is not always clear what is compared.
  2. Looking at the obtained data, it is hard to see some pattern of antioxidative and inflammatory status that resulted exclusively from 2S-hesperidin treatment. Effects have also been detected in placebo group, but the authors accentuated findings observed in 2S-hesperidin group.
  3. In the Abstract, the authors have written: „2S-Hesperidin has shown an antioxidant and anti-inflammatory effect in animal studies, but so far no one has studied this effect in humans. „ – this is not correct as authors have published 2 papers related to the effects of 2S-hesperidin in amateur cyclists.

Antioxidant and anti-inflammatory properties of hesperidin are described through the literature. The authors wrote in the Introduction that hesperidin reduces SOD and GSSG (hence, these effects are already known).  Similarly, on page 2, lines 45-47, they wrote: „Moreover, the intake of hesperidin (in orange juice) has shown to modulate leukocyte gene expression, boosting its antioxidant and inflammatory profile, showing therefore a nutrigenomic effect [6].“ - The novelty of the work (related to antioxidative and anti-inflammatory effects of hesperidin) is questionable.

  1. The conclusions are overstated.

At baseline, oxidative status improved (↓TBARS) after intervention with 2S-hesperirin and compared to placebo. In addition, significant improvements in la antioxidant capacity (↑SOD) after maximal exercise, and inflammatory status after the acute recovery phase (↓MCP1) were found in 2S-hesperidin group and compared to placebo (baseline and Post-REC).“

If I understood correctly, p for TBARS is 0.052, this is not smaller than 0.05, hence it is not significant and should not be emphasized as the evidence of the improvement of the oxidative status.

On the other hand, changes observed in placebo group were neglected. For example, changes of IL6 and TNFα have been found in placebo group as well as decrease in MCP1 and increase in SOD1 (evaluated as AUC). Results in placebo group must be discussed and correctly compared with the 2S-hesperidin group.

  1. Page 14, line 408 – The authors wrote: „In our study, exercise decreased intra-group Post-PMAX GSSG levels when comparing pre- and post-intervention in both groups“ - unclear (p in hesperidin group is 0.058 in Post Pmax, hence GSSG is not decreased), whereas in placebo group p=0.049 which is not discussed.

The authors wrote that „The main advantage of incorporating the AUC in this study is that it allows us to precisely define the duration and magnitude of the variable being evaluated, which cannot be done in a point-by-point comparison [39]. Based on AUC, both GSH and GSSG were depleted, and decrease in GSH is more prominent in 2S-hesperidin group. Decreased levels of GSH may indicate reduced ability to cope with ROS moieties. Please comment.

Page 15, line 458:  „In this study, cytokine levels increased during the first and second rectangular test from baseline to Post-PMAX in both groups, but there were different trends from Post-PMAX to Post-REC in second rectangular test (↓2S-hesperidin and ↑placebo).” – this conclusion is based on statistical analysis, or just means are compared?

Supplementary file 1 has been published – Table 2 in Nutrients https://www.mdpi.com/2072-6643/12/12/3911/htm

Table 1 and Table 2 have been published in Nutrients https://www.mdpi.com/2072-6643/12/12/3911/htm

Page 1, line 39 – may be found

Page 2, line 51 – radical scattering or radical scavenging

Page 3, line 93 - (HO1) – full name is missing

Page 3, line 97 - in amateur cyclists before, after of rectangular test and resting phase – unclear

Abbreviation RPE is not explained in Table 1

Table 4 – CRP instead of CRP

Page 5, line 158 – RER?

NF-κB instead of NF-κβ

Author Response

Reviewer 2

Following 8-week supplementation with 2S-hesperidin, the authors measured some antioxidative and inflammatory parameters in amateur cyclists after exercise until exhaustion. In my opinion, the manuscript must be improved before considering publication.

Response:  We thank the reviewer for their constructive and helpful feedback on our manuscript. We have replied to each specific comment in the section below and have introduced the corresponding edits into the manuscript using Word’s track changes.

  1. 1. The manuscript is very hard to follow. English must be improved (professional assistance is suggested). The design of the study, as well as the statistical analysis, are not described in an understandable manner for an average reader. There are many groups and time points, and is not always clear what is compared.

Response:  Following your suggestion, we have modified the study design and the statistics section. If there are still any parts of the study design or statistical results that are not well understood, please let us know. Thank you for your comment.

  1. 2. Looking at the obtained data, it is hard to see some pattern of antioxidative and inflammatory status that resulted exclusively from 2S-hesperidin treatment. Effects have also been detected in placebo group, but the authors accentuated findings observed in 2S-hesperidin group.

Response:  We understand that this is a lot of information, but in order to better understand the results, we have introduced both tables and figures, which make it easier to understand the changes. In the discussion, we have discussed the placebo results.

  1. 3. In the Abstract, the authors have written: „2S-Hesperidin has shown an antioxidant and anti-inflammatory effect in animal studies, but so far no one has studied this effect in humans. „ – this is not correct as authors have published 2 papers related to the effects of 2S-hesperidin in amateur cyclists.

Response: We refer to the chronic intake of 2S-hespridine in this sentence, but we understand your suggestion and have specified it in the manuscript in lines 15-16. The first paper we published was on acute 2S-hesperidin intake and evaluated antioxidant markers. The second paper published deals with chronic 2S-hesperidin intake and sports performance and does not discuss any antioxidant and inflammatory markers.

Antioxidant and anti-inflammatory properties of hesperidin are described through the literature. The authors wrote in the Introduction that hesperidin reduces SOD and GSSG (hence, these effects are already known).  Similarly, on page 2, lines 45-47, they wrote: „Moreover, the intake of hesperidin (in orange juice) has shown to modulate leukocyte gene expression, boosting its antioxidant and inflammatory profile, showing therefore a nutrigenomic effect [6].“ - The novelty of the work (related to antioxidative and anti-inflammatory effects of hesperidin) is questionable.

Response: The 2S-hesperidin content in our supplement is higher and in orange juice is different. But it should be noted that orange juice contains other flavonoids which also have antioxidant and anti-inflammatory activity, such as naringenin, eriocitrin, etc., and may therefore have a synergistic effect with 2S-hesperidin. The novelty of our study is that we used a very rarely marketed form of hesperidin (2S-hesperidin) and, in addition, participants chronically took the supplement (8 weeks) and were without pathologies (cyclists).

  1. 4. The conclusions are overstated.

„At baseline, oxidative status improved (↓TBARS) after intervention with 2S-hesperirin and compared to placebo. In addition, significant improvements in la antioxidant capacity (↑SOD) after maximal exercise, and inflammatory status after the acute recovery phase (↓MCP1) were found in 2S-hesperidin group and compared to placebo (baseline and Post-REC).“

If I understood correctly, p for TBARS is 0.052, this is not smaller than 0.05, hence it is not significant and should not be emphasized as the evidence of the improvement of the oxidative status.

Response: Although the p-value is not exactly equal to or below 0.05 because of the third decimal place, we decided that if this number were expressed with 2 decimal places, it would be equal to 0.05 and with moderate effect (>0.5), and we believe that it can be interpreted as a significant effect. Following your suggestion, we have modified the sentence. Lines 386-389.

On the other hand, changes observed in placebo group were neglected. For example, changes of IL6 and TNFα have been found in placebo group as well as decrease in MCP1 and increase in SOD1 (evaluated as AUC). Results in placebo group must be discussed and correctly compared with the 2S-hesperidin group.

Response: Following your suggestion, we have introduced 2 more paragraphs into the discussion, discussing the changes in IL6 and TNFα in placebo. Lines 495-503. 

Response: The changes in the AUC of SOD in placebo are not discussed because they are similar to those of 2S-hesperidin, and it would lengthen the discussion in giving possible explanations as to why the 2 groups had a similar effect in relation to AUC.

Response: Following your suggestion, we have introduced 2 more paragraphs into the discussion, discussing the changes in MCP1 in placebo. Lines 563-567.

Page 14, line 408 – The authors wrote: „In our study, exercise decreased intra-group Post-PMAX GSSG levels when comparing pre- and post-intervention in both groups“ - unclear (p in hesperidin group is 0.058 in Post Pmax, hence GSSG is not decreased), whereas in placebo group p=0.049 which is not discussed.

Response: Following your suggestion, we have deleted the sentence and introduced a new sentence in relation to the GSH and GSSG changes. Lines 441-446.

The authors wrote that „The main advantage of incorporating the AUC in this study is that it allows us to precisely define the duration and magnitude of the variable being evaluated, which cannot be done in a point-by-point comparison [39]. Based on AUC, both GSH and GSSG were depleted, and decrease in GSH is more prominent in 2S-hesperidin group. Decreased levels of GSH may indicate reduced ability to cope with ROS moieties. Please comment.

Response: As mentioned in line 441-443, this decrease may be due to a decrease in the volume and intensity of the cyclists' training in the period in which the study was conducted compared to other periods of the cycling season, ref 39. Therefore, this decrease in both groups is due to a lower exposure to high levels of free radicals leading to maladaptation in the glutathione antioxidant system. It would have been interesting to also measure glutathione peroxidase and reductase, but this was not possible due to budget constraints.

Page 15, line 458:  „In this study, cytokine levels increased during the first and second rectangular test from baseline to Post-PMAX in both groups, but there were different trends from Post-PMAX to Post-REC in second rectangular test (↓2S-hesperidin and ↑placebo).” – this conclusion is based on statistical analysis, or just means are compared?

Response: These changes were expressed from the means of each group and points in time, but there was a reference to IL6, which has been modified in the manuscript in line 485.

Supplementary file 1 has been published – Table 2 in Nutrients https://www.mdpi.com/2072-6643/12/12/3911/htm

Table 1 and Table 2 have been published in Nutrients https://www.mdpi.com/2072-6643/12/12/3911/htm

Response: Both table 1 and 2 and supplementary file 1 are published in another paper in "Nutrients", as both papers are part of the same project and the measurements were performed at the same time. The data in tables 1 and 2, are descriptive data for the sample and not for repeated results. If you review the clinical trial register, you will see that all of the measurements made in this, which encompasses many variables, could not be included into a single paper. If you consider it appropriate, please let us know precisely what to do with these files.

Page 1, line 39 – may be found

Response: Amended

Page 2, line 51 – radical scattering or radical scavenging

Response: Amended

Page 3, line 93 - (HO1) – full name is missing

Response: Amended

Page 3, line 97 - in amateur cyclists before, after of rectangular test and resting phase – unclear

Response: Amended

Abbreviation RPE is not explained in Table 1

Response: Amended

Table 4 – CRP instead of CRP

Response: Amended

Page 5, line 158 – RER?

NF-κB instead of NF-κβ

Response: Amended

Author comment: We appreciate all the comments made on our manuscript, which helped improve it’s quality.

Round 2

Reviewer 2 Report

Regarding the revised version of the manuscript, I still have some concerns. I suggest authors to rewrite parts of abstract/discussion/conclusions based on the comments given below.

  1. As suggested, the authors have introduced new sentences regarding the effects that were observed in the placebo group, but still my impression is that conclusions about the effects of hesperidin are overstated.

The authors monitored several parameters of oxidative and inflammatory status. Regarding oxidative stress status the authors investigated levels of TBARS, catalase, superoxide dismutase (SOD), GSH and GSSG, and HO1. SOD was increased in hesperidin group (the same was observed in placebo), whereas GSSH was decreased in one time point (the same was observed in placebo group for another time point). GSH was depleted (the same was observed in placebo group), whereas increase in TBARS almost rich statistical significance (p=0.052) and was found only in hesperidin group. This may indicate enhanced production of ROS in hesperidin group. Change in HO-1 was observed in placebo group, but not in hesperidin group. Hence, based on these findings, it is not correct to interpret that treatment with hesperidin seems to improve antioxidant status (abstract). Furthermore, looking at Figure 3, it seems that observed effects are of statistical, but not of physiological importance (differences between columns are very, very small).

When considering inflammatory status, four parameters were monitored: IL-6, TNF-α, MCP-1 and CRP. Again, only MCP-1 was reduced in hesperidin group, whereas IL-6, TNF-α and MCP-1 were reduced in placebo group at some points. However, the conclusion drawn is that hesperidin seems to improve the inflammatory status (abstract), although even better effects were observed in placebo group.  

The study is interesting and valuable, but I am not convinced that hesperidin significantly affected antioxidative and inflammatory status in amateur cyclists - in my opinion, this needs to be pointed out. 

  1. Almost all grammatical errors from the original manuscript are still present in the revised version (some examples: line 25: a decreased; line 29: seem; line 37: the molecules; line 82: ROS are an………). The manuscript is still hard to follow and understand.

Minor points:

  • abstract is repetitive
  • Discussion – lines 398-412 – speculation about hesperidin-induced Nrf2 activation based on SOD increase (SOD increase was also observed in placebo group) – could be shortened
  • line 523 – E.coli is not harmful bacteria in the gut
  • nuclear factor kappa B (NF-Ò¡B) not nuclear factor kappa beta (NF-Ò¡β)
  • MCP1 values in Table 4 are not organized well

Author Response

Regarding the revised version of the manuscript, I still have some concerns. I suggest authors to rewrite parts of abstract/discussion/conclusions based on the comments given below.

Response:  We thank the reviewer for their constructive and helpful feedback on our manuscript. We have replied to each specific comment in the section below and have introduced the corresponding edits in the manuscript using Word’s track changes.

As suggested, the authors have introduced new sentences regarding the effects that were observed in the placebo group, but still my impression is that conclusions about the effects of hesperidin are overstated.

Response: Our intention was to describe what is happening to the antioxidant and inflammatory markers. So, we don't believe the conclusions are overstates. First of all, in our view, placebo does not improve anything, rather it would be physical activity that improves, worsens or does not affect a certain marker at a certain point in time. And secondly, an intervention with a substance with demonstrated bioactive properties, if it produces a change with respect to the placebo, we can say that it improves or does not improve a certain marker at a certain point in the test. Therefore, we think that the placebo cannot improve anything because it does not have a bioactive compound.

The authors monitored several parameters of oxidative and inflammatory status. Regarding oxidative stress status the authors investigated levels of TBARS, catalase, superoxide dismutase (SOD), GSH and GSSG, and HO1. SOD was increased in hesperidin group (the same was observed in placebo), whereas GSSH was decreased in one time point (the same was observed in placebo group for another time point). GSH was depleted (the same was observed in placebo group), whereas increase in TBARS almost rich statistical significance (p=0.052) and was found only in hesperidin group. This may indicate enhanced production of ROS in hesperidin group. Change in HO-1 was observed in placebo group, but not in hesperidin group. Hence, based on these findings, it is not correct to interpret that treatment with hesperidin seems to improve antioxidant status (abstract). Furthermore, looking at Figure 3, it seems that observed effects are of statistical, but not of physiological importance (differences between columns are very, very small).

Response: In response to your suggestions, we have removed the last sentence from the abstract so that readers are not confused. But we consider that the changes in SOD are not similar or of the same magnitude in 2S-hesperidin as in placebo, as 2S-hesperidin had significant differences in both Post-PMAX, Post-REC and AUC with moderate to large size effects, whereas placebo only had significant differences in baseline and AUC. In addition, improvements in post-exercise antioxidant markers are ideal, as they would help improve recovery.

Response: You indicate that there were the same changes in GSSG in both 2S-hesperidin and placebo, but you have missed the significant change in GSSG AUC in 2S-hesperidin, which as discussed in the manuscript better represents an overall rectangular test effect.  In addition, in our previous study, the 2S-hesperidin group improved submaximal and maximal performance in the incremental test have to specify that together with the rectangular test measurements, thus increasing their capacity to produce power at high intensities (https://doi.org/10.3390/nu12123911). These results suggest that the improvements in SOD after Post-PMAX rectangular test could be related to improvements in power production at high intensities, which could justify lower levels of GSH after Post-PMAX in 2S-hesperidin. However, lower levels of GSSG post-intervention were observed, which indicates lower level of oxidative stress. Interestingly, the placebo group did not show any change in power produced at high intensities and SOD Post-PMAX.  

Response: We agree that TBARS has higher levels at baseline pre post-intervention in 2S-hesperidin without being significant, but there is a trend. But we must also take into account the lower values in the 2S-hesperidin group when comparing the post-intervention values at baseline between groups, which you do not mention in the previous paragraph, where you state the background. It should be noted that GSSG is a marker of oxidative stress. 

Response: We have included a sentence in the article specifying that although there is no clear pattern of improvement in antioxidant markers, there is an improvement in certain components (↑SOD, ↓GSSG) of the endogenous antioxidant system measured in this study and at key times during recovery. However, further studies are needed to provide clarity on this issue. Lines 462-465.

Response: Two isoforms have been identified that are separate gene products: heme oxygenase-1 (HO-1), the inducible form (also known as heat shock protein 32), and HO-2, the constitutive form. Recently, evidence of a second constitutive isoform (HO-3) has been found. The role of heme oxygenase in different tissues has not, as yet, been fully characterized, but it is becoming evident that it is involved in a variety of cellular regulatory and protective mechanisms. HO-1 has been shown to be protective against ischemia-reperfusion and free radical damage in a number of tissues. Several possible mechanisms for this protection have been proposed: 1) elimination of the pro-oxidant free heme could decrease hydroxyl radical formation, thereby reducing cellular damage; 2) biliverdin and the reduced product bilirubin are both potent free radical scavengers with antioxidant properties; and 3) carbon monoxide, as a vasodilator, could play an important role in maintaining cellular integrity and function under certain conditions such as ischemia. The relative importance of each of these factors in cytoprotection under different pathophysiological conditions has not been established but is currently under investigation.  Myoglobin, a potential source of free heme, is released from skeletal muscle during severe ischemia reperfusion injury and is implicated in the pathogenesis of acute renal failure. The precise mechanism of this is unclear; however, both iron released from heme and H2O2 generation mediated by heme have been proposed as mediators of the damage. Induction of HO-1 in the kidney has been shown to protect against this form of renal injury. Whether a similar mechanism of injury occurs in the muscle itself during ischemia-reperfusion and whether HO-1 has a protective role in this tissue are not known. The HO-1 isoform is extremely sensitive to a variety of agents that cause oxidative stress, including heat shock, ischemia, hypoxia, and endotoxin. It is also inducible by hemin and the free radical nitric oxide (NO). HO-1 is induced in rat skeletal muscle following exhaustive exercise and electrical stimulation. 10.1152/ajpcell.1998.275.4.C1087

Based on the above paragraph we can interpret the induction in the acute recovery phase of HO-1 in placebo as being due to an exercise-mediated increase in a component of the oxidative system.  Therefore, this is a physiological effect of high intensity exercise and possibly having an attenuation of the increase of HO-1 in the group that ingested 2S-hesperidin shows an attenuation of oxidative stress in the acute phase of recovery.

Thank you for your comment.

When considering inflammatory status, four parameters were monitored: IL-6, TNF-α, MCP-1 and CRP. Again, only MCP-1 was reduced in hesperidin group, whereas IL-6, TNF-α and MCP-1 were reduced in placebo group at some points. However, the conclusion drawn is that hesperidin seems to improve the inflammatory status (abstract), although even better effects were observed in placebo group.  

Response: Following your suggestions, we have removed the last sentence from the abstract. We have included a sentence in the discussion to try not to confuse readers. Lines 547-550. Thank you for your comment

The study is interesting and valuable, but I am not convinced that hesperidin significantly affected antioxidative and inflammatory status in amateur cyclists - in my opinion, this needs to be pointed out. 

Response: We understand your position, but based on the results of this research, to say that there is no effect would not be telling the truth either. Therefore, we maintain that there are some benefits at some points in the rectangular test on some of the antioxidant and inflammatory markers in this study.

Almost all grammatical errors from the original manuscript are still present in the revised version (some examples: line 25: a decreased; line 29: seem; line 37: the molecules; line 82: ROS are an………). The manuscript is still hard to follow and understand.

Response: Following your suggestions, we have modified the grammatical errors. Thank you for your comment.

Minor points:

abstract is repetitive

Discussion – lines 398-412 – speculation about hesperidin-induced Nrf2 activation based on SOD increase (SOD increase was also observed in placebo group) – could be shortened

Response: Following your suggestion, we have shortened this paragraph.

line 523 – E.coli is not harmful bacteria in the gut

Response: Based on the conclusions of several studies that E.coli is the cause of various pathologies, we determined the acceptance that the intestinal presence of this bacterium is not beneficial. Doi.10.1093/femsre/fuw005. Doi.10.1016/j.cmi.2014.09.021.

nuclear factor kappa B (NF-Ò¡B) not nuclear factor kappa beta (NF-Ò¡β)

Response: Amended.

MCP1 values in Table 4 are not organized well

Response: We do not understand this statement, as the data are organized as in the previous variables, by rectangular test point and pre- and post-intervention, with the groups separated. Could you be more specific in pointing out our error?

Author comment: We appreciate all the comments made on our manuscript, which helped improve it’s quality.

Round 3

Reviewer 2 Report

I thank authors for their detailed replies to my concerns. Values for MCP-1 in Table 4 for are not well aligned.